# S100A10 Promotes Pancreatic Ductal Adenocarcinoma Cells Proliferation, Migration and Adhesion through JNK/LAMB3-LAMC2 Axis

**DOI:** 10.3390/cancers15010202

**Published:** 2022-12-29

**Authors:** Hai Lin, Pengfei Yang, Bixiang Li, Yue Chang, Yutong Chen, Yaning Li, Kecheng Liu, Xinyue Liang, Tianliang Chen, Yalan Dai, Wenzheng Pang, Linjuan Zeng

**Affiliations:** 1Cancer Center of the Fifth Affiliated Hospital, Sun Yat-sen University, Zhuhai 519000, China; 2Department of Pathology, The Fifth Affiliated Hospital, Sun Yat-sen University, Zhuhai 519000, China; 3Guangdong Provincial Key Laboratory of Biomedical Imaging, The Fifth Affiliated Hospital, Sun Yat-sen University, Zhuhai 519000, China

**Keywords:** pancreatic ductal adenocarcinoma (PDAC), S100A10, LAMB3, LAMC2, proliferation, migration, adhesion

## Abstract

**Simple Summary:**

Despite the advance in therapeutic strategy, the prognosis of pancreatic ductal adenocarcinoma (PDAC) is still unsatisfactory, with a 5-year survival rate of less than 9%. As a member of the S100 protein family, S100A10 has been identified as an oncogene in multiple cancers. In PDAC, S100A10 has been reported to be not only up-regulated in tumor tissues but also associated with survival outcomes. However, the specific function of S100A10 in PDAC is still unknown. Here, we suggest that S100A10 promotes PDAC cells proliferation, migration, and adhesion by activating JNK/LAMB3-LAMC2 axis in vitro and accelerates pancreatic tumor growth in vivo, indicating that S100A10 may be a potential therapeutic target for PDAC patients.

**Abstract:**

Pancreatic ductal adenocarcinoma (PDAC) is one of the most aggressive tumors, characterized by diagnosis at an advanced stage and a poor prognosis. As a member of the S100 protein family, S100A10 regulates multiple biological functions related to cancer progression and metastasis. However, the role of S100A10 in PDAC is still not completely elucidated. In this study, we reported that S100A10 was significantly up-regulated in PDAC tissue and associated with a poor prognosis by integrated bioinformatic analysis and human PDAC tissue samples. In vitro, down-regulation of S100A10 reduced the proliferation, migration, and adhesion of PDAC cell lines, whereas up-regulation of S100A10 showed the opposite effect. Furthermore, LAMB3 was proved to be activated by S100A10 using RNA-sequencing and western blotting. The effect of LAMB3 on the proliferation, migration, and adhesion of PDAC cells was similar to that of S100A10. Up-regulation or down-regulation of LAMB3 could reverse the corresponding effect of S100A10. Moreover, we validated S100A10 activates LAMB3 through the JNK pathway, and LAMB3 was further proved to interact with LAMC2. Mice-bearing orthotopic pancreatic tumors showed that S100A10 knocked-down PANC-1 cells had a smaller tumor size than the control group. In conclusion, S100A10 promotes PDAC cells proliferation, migration, and adhesion through JNK/LAMB3-LAMC2 axis.

## 1. Introduction

Pancreatic ductal adenocarcinoma (PDAC) is a common digestive system malignancy with high invasiveness and metastasis. The latest report on cancer showed that both the incidence and mortality of PDAC had been increasing during 1990–2017 worldwide [1], and it has been the third leading cause of cancer-related deaths in the United States [2]. Despite the development of preventive and therapeutic strategies, the prognosis of PDAC is still unsatisfactory, with a 5-year survival rate of less than 9%. It is attributed to the diagnosis at advanced stages in most PDAC patients and the high recurrence rate after resection, as well as the resistance to drug therapy [3,4]. Therefore, it is imperative to elucidate the molecular mechanisms of the occurrence and progression of PDAC to develop novel therapeutic strategies for improving their survival outcomes.

The S100 calcium-binding protein family (S100s) is a class of small molecular proteins expressed only in vertebrates and consists of 21 family members with a similar EF-hand motif structure [5]. As a unique member of S100s, S100A10 has similar EF-hand motifs, but its key amino acids in the motifs have been substituted or deleted, which maintains it in a continuously activated state and is insensitive to Ca^2+^ [6]. Additionally, S100A10 often interacts with annexin A2 (ANXA2) to form an ANXA2-S100A10 complex to play various biological regulatory roles [7]. Up to now, S100A10 has been reported to be up-regulated in several cancers, including lung cancer [8], gall bladder adenocarcinoma [9], colorectal adenocarcinoma [10], and hepatocellular carcinoma [11], which correlated with worse survival outcomes. The significant up-regulation of S100A10 has also been revealed in PDAC tissues, compared with that in normal pancreas tissues or pancreatitis tissues [12,13]. Moreover, it is demonstrated that S100A10 is regulated by the oncogene KRAS in PDAC [14]. However, little is known about the functional roles of S100A10 in PDAC.

In this study, we compared the expression of S100A10 between PDAC tumor tissues and pancreas non-tumor tissues and further analyzed the correlation between S100A10 expression and clinicopathological characteristics to evaluate its prognostic value for PDAC patients. Furthermore, a series of in vitro functional experiments were performed to investigate the role of S100A10 in the progression of PDAC and identify the genes activated by S100A10. An orthotopic xenograft model was also performed to evaluate the role of S100A10 on pancreatic tumor growth in vivo.

## 2. Materials and Methods

### 2.1. Oncomine Database Analysis

The transcriptional expressions of S100s were analyzed in twenty types of human cancers, including PDAC and corresponding normal tissues in Oncomine database (www.Oncomine.org, accessed on 17 October 2021), an online public database that integrates the DNA and RNA sequencing data from multiple human cancers. The difference in mRNA expression was compared by Student’s *t*-test (cutoff *p*-value: 0.01, cutoff fold-change: 1.5).

### 2.2. GEPIA Database Analysis

The mRNA expressions of S100s were compared between 179 PDAC tissues and 171 normal pancreas tissues in Gene Expression Profiling Interactive Analysis database (GEPIA, http://gepia.cancer-pku.cn/, accessed on 21 October 2021), based on the data derived from The Cancer Genome Atlas (TCGA) and Genotype Tissue Expression project (GTEx). The cutoff of *p*-value and fold-change were 0.01 and 2, respectively. The correlation analyses between two genes in PDAC were also performed in GEPIA database by using Pearson correlation coefficient, with *p*-value < 0.05 defined as statistically significant.

### 2.3. Kaplan–Meier Plotter Database Analysis

According to their median expression levels in PDAC tissues, the mRNA expressions of S100s were divided into two groups (Low expression group vs. High expression group) and further used to make survival analyses to overall survival (OS) and recurrence-free survival (RFS) for PDAC patients in Kaplan–Meier plotter database (https://kmplot.com/, accessed on 21 October 2021), an online public database, which can be used to make survival analysis on the expression of 54,675 genes in 21 types of human cancers. The logrank *p* < 0.05 is defined as statistically significant.

### 2.4. UALCAN Database Analysis

UALCAN database (http://ualcan.path.uab.edu/, accessed on 21 October 2021) is an interactive website providing analysis of cancer OMICS data. The correlation analyses between the mRNA expression of S100A2/10 in PDAC tissues and clinicopathological characteristics, including grade and stage, were performed in UALCAN database based on the data derived from The Cancer Genome Atlas (TCGA). The comparison was made by Student’s *t*-test, with *p*-value < 0.05 regarded as statistically significant.

### 2.5. Human Protein Atlas Database (HPA) Analysis

The Human Protein Atlas database (HPA, http://www.proteinatlas.org, accessed on 25 October 2021) is an online public website, which provides data on the proteome from 17 different human cancer in ten aspects, including The Tissue Atlas, The Brain Atlas, The Single Cell Type Atlas, The Cell Type Atlas, The Pathology Atlas and so on. To evaluate differences in S100A10, ITGA2, LAMB3, and LAMC2 expression at the protein level, immunohistochemistry images of S100A10, ITGA2, LAMB3, and LAMC2 protein expressions in normal pancreas tissues and PDAC tissues, were downloaded from The Tissue Atlas and The Pathology Atlas in HPA, respectively.

### 2.6. cBioPortal Database Analysis

cBioPortal database (http://www.cbioportal.org/, accessed on 31 October 2021), an online tool that integrates multidimensional cancer genomics data, was used to acquire the genes significantly associated with S100A10 in PDAC tissues in the QCMG project (Nature 2016, 456 samples) and TCGA project (PanCancer Atlas, 184 samples), with *q*-value < 0.05 and *q*-value < 0.01 regarded as statistically significant, respectively.

### 2.7. DEGs Analysis in GEO and UCSC Xena Database

In the Gene Expression Omnibus database (GEO, http://www.ncbi.nlm.nih.gov/geo, accessed on 31 October 2021), the public microarray dataset GSE62452 was analyzed with an online tool GEO2R to acquire the significantly different expression genes (DEGs) between 69 PDAC samples and 61 adjacent non-tumor tissue samples (*p*adj-value < 0.05, |log2-fold change| > 1). The DEGSeq2 standardized transcriptional data in the TCGA-TARGET-GTEx cohort were downloaded from the public online website: UCSC Xena (https://xenabrowser.net/, accessed on 31 October 2021). Statistical software R (version 4.1.0, https://cran.r-project.org/, accessed on 10 September 2021) was used to conduct significance analysis of DEGs between 179 PDAC samples and 171 normal pancreas samples. DEGSeq2 package (version 1.32.0, http://www.bioconductor.org/, accessed on 10 September 2021) was used to select significant DEGs, where the standard was *p*adj-value < 0.05 and |log2-fold change| > 2.

### 2.8. Gene Ontology (GO), Kyoto Encyclopedia of Genes and Genomes (KEGG) Analysis

After the intersection of the genes identified to be significantly associated with S100A10 in PDAC in cBioPortal database analysis and the significant DEGs identified in GEO and UCSC Xena database analyses, 55 genes were acquired. Survival analyses for these 55 genes were further performed in GEPIA database, and only 38 genes were identified to be significantly associated with the OS for PDAC patients (logrank *p*-value < 0.05). After gene ID conversion, these 38 genes were used for Gene ontology (GO) term enrichment analysis (including biological process, cellular component, and molecular function) and KEGG pathway enrichment analysis with the Statistical software R (version 4.1.0, https://cran.r-project.org/, accessed on 10 September 2021) and cluster Profile package (version 4.0.5, http://www.bioconductor.org/, accessed on 10 September 2021), where the standard was *p*-value < 0.05.

### 2.9. Cells and Reagents

Six PDAC cell lines (including PANC-1, CFPAC-1, HPAF II, MIAPaCa-2, AsPC-1 and Bx-PC3) were obtained from the American Type Culture Collection (ATCC, Manassas, VA, USA) and cultured in Dulbecco’s modified Eagle’s medium (DMEM) (Gibco, Thermo Fisher Scientific, Waltham, MA, USA), supplemented with 10% fetal bovine serum (Gibco, Thermo Fisher Scientific, Waltham, MA, USA) at 37 °C in a humidified atmosphere containing 5% CO_2_. The JNK inhibitor SP600125 was purchased from Selleck (Houston, TX, USA).

### 2.10. Transient Transfection, Lentiviral Transduction and Generation of Stable Cell Lines

Two siRNAs against S100A10 (siS100A10-1, siS100A10-2), one siRNA against LAMB3 (siLAMB3), and corresponding control siRNAs (siNC) were purchased from RIBOBIO (Guangzhou, China). A plasmid containing full-length S100A10 cDNA (NM_002966) (pS100A10) and another plasmid containing full-length LAMB3 cDNA (NM_000228.3) (pLAMB3) as well as corresponding control plasmids (pNC) were purchased from MiaoLing (Wuhan, China). PDAC cells were seeded into 6-well plates at 5 × 10^5^ cells/well density and incubated 24 h to reach 50–60% confluence for transfection. Transfections were performed with a Lipofectamine-3000 transfection kit (Thermo Fisher Scientific, Waltham, MA, USA), according to the manufacturer’s protocol. After transfection for appropriate time, cells were harvested and used for further in vitro experiments. After transfection with pNC or pS100A10 for 24 h, PDAC cells were exposed to a JNK inhibitor (SP600125) at 60 umol concentration for 48 h, or transfected with a siRNA against JNK1 (siJNK1) or control siRNA (siNC) for 48 h, followed by western blotting analysis to measure the protein expressions of interesting genes. The target sequences used in the siRNA were summarized in Appendix A. S100A10 stably knocked-down (shS100A10) and negative control (shNC) PANC-1 and AsPC-1 cells were generated by using lentiviral constructs expressing shS100A10 and shNC, which were designed by Genechem (Shanghai, China). Briefly, PANC-1 and AsPC-1 cells were seeded into 6-well plates at 2 × 10^5^ cells/well density and incubated overnight, followed by lentiviral infection (MOI = 10) for 24 h. Subsequently, the shNC and shS100A10 PANC-1 and AsPC-1 cells were selected with puromycin (MP, Santa Ana, CA, USA) at 5 μg/mL concentration for 7 days.

### 2.11. RNA Sequencing (RANseq) and Database for Annotation, Visualization and Integrated Discovery (DAVID) Analysis

PANC-1 cells were seeded into 6-well plates at 5 × 10^5^ cells/well density and incubated 24 h to reach 50–60% confluence, followed by transfected with a siRNA against S100A10 (siS100A10-1) or control siRNA (siNC), a plasmid containing full-length S100A10 cDNA (pS100A10) or control plasmid (pNC) for 72 h. The transfection concentrations of siRNAs and plasmids were 100 nM and 2.5 ng/uL, respectively. Subsequently, the cells were collected and washed 2 times with 1× PBS. A total of 1ml Trizol was added to each sample (siNC, siS100A10-1, pNC, and pS100A10) and sent to GENEWIZ Corporation (Suzhou, China) for RNAseq. RNAseq was performed by using the Illumina HiSeq X platform. Briefly, total RNA was isolated from each sample, followed by quality control. Subsequently, cDNA was synthesized and further amplified and purified. Htseq Software (V 0.6.1, https://anaconda.org/bcbio/htseq, accessed on 10 October 2021) and Fragments Per Kilo bases per Million reads (FPKM) method were used to calculate gene expression. When analyzing the RNAseq data (read count) of 4 groups (including siNC, siS100A10-1, pNC, and pS100A10), we first filtered the low-expressed genes (read count ≤ 25) and further set the screening thresholds for each gene as: “siS100A10-1/siNC ≤ 0.5 and pS100A10/pNC ≥ 2” or “siS100A10-1/siNC ≥ 2 and pS100A10/pNC ≤ 0.5”. One-hundred-thirty-four genes were identified and further used for KEGG pathway enrichment analysis in the Database for Annotation, Visualization and Integrated Discovery (DAVID) (https://david.ncifcrf.gov/, accessed on 24 January 2022), with *p*-value < 0.05 defined as statistical significance.

### 2.12. The Human PDAC Tissue Samples Analysis and Immunohistochemistry (IHC)

PDAC specimens (including 43 PDAC tissues and 31 adjacent non-tumor tissues) were collected from patients who received an operation at the Fifth Affiliated Hospital of Sun Yat-sen University from October 2004 to May 2020. The clinicopathological characteristics of these 43 PDAC patients were summarized in Appendix A. The protocol was approved by the Medical Ethics Committee of the Fifth Affiliated Hospital of Sun Yat-sen University, and the informed consent was written. These specimens were embedded with paraffin and subsequently used for immunohistochemistry. The immunohistochemistry was performed by using an IHC staining kit (SA1022, Boster, Wuhan, China) according to the manufactory protocol on paraffin-embedded tissue. In brief, after deparaffinization, rehydration, antigen retrieval, endogenous peroxidase blocking, and 5% BSA blocking, the specimens were incubated with primary antibodies overnight at 4 °C, and subsequently biotinylated with a secondary antibody for 1 h at 37 °C, followed by visualized using a DAB reagent (KIT-9710-6 ml, MXB, Fuzhou, China) and counterstained with Mayer’s hematoxylin (DH0005, LEAGENE, Beijing, China) for 10 min. The primary antibodies against S100A10, LAMB3, and ITGA2 were used at 1/100, 1/300, and 1/500 dilution in 1×PBS, respectively. Finally, two independent pathologists who were not aware of the clinicopathological data of 43 PDAC patients interpreted the results. The immunoreactive score (IRS) was calculated by the quantity and intensity of the staining. The number of positive cells was scored as follows: 0 points (<10% cell staining), 1 point (10–25% cell staining), 2 points (26–75% cell staining) and 3 points (>75% cell staining). The intensity of staining was calculated as follows: 0 points (negative), 1 point (weak), 2 points (moderate), and 3 points (strong). Images were captured by a microscope (Leica, Wetzlar, Hessen, Germany) at 50×, 200× or 400× magnification. Antibodies’ information for IHC is listed in Appendix A.

### 2.13. Real-Time Quantitative PCR (RT-qPCR) for Measurement of Gene Expression

Total RNA was extracted by using an RNA-Quick Purification Kit (RN001, ES Science, Beijing, China) according to the manufacturer’s protocol. Subsequently, cDNA was synthesized from total RNA by using a PrimeScript™ RT Master Mix reagent kit (TaKaRa, Beijing, China). Real-time quantitative polymerase chain reaction (RT-qPCR) was performed to compare the relative mRNA expression level of interesting genes by using Hieff UNICON^®^ Univeral Blue qPCR SYBR Green Master Mix (11184ES08, YEASEN, Shanghai, China) on the Agilent AriaMx Real-time PCR System (Agilent, Palo alto, California, USA). The sequences of primers for RT-qPCR are listed in Appendix A.

### 2.14. Western Blotting

Protein samples were prepared by using RIPA lysis buffers (CWBIO, Jiangsu, China) containing 1× protease inhibitors (Selleck, Houston, TX, USA) and 1× phosphatase inhibitors (Selleck, Houston, TX, USA). Subsequently, western blotting was performed to detect the protein expression levels of interesting genes. Briefly, SDS-PAGE on 10% or 15% gel was used for protein separation, and the separated proteins were transferred to a PVDF membrane (Merck millipore, Billerica, MA, USA) and blocked with 5% non-fat milk in 1× PBST for 1 h, followed by incubated with primary antibodies overnight at 4 °C. After the incubation of HRP-conjugated secondary antibodies (1:5000~10,000) in 1× PBST at room temperature for 1 h, the signals of protein levels were detected with a Chemistar ECL Western Blotting Substrate (180–5001, Tanon, Shanghai, China) by using a Tanon-5200 multi-imaging system (Tanon, Shanghai, China). The primary antibodies against S100A10 (1/1000), LAMB3 (1/1000), ITGA2 (1/2500), β-actin (1/2500), Flag (1/1000), p105/NF-kB (1/1000), p-p105/NF-kB (1/1000), p65/NF-kB (1/1000), p-p65/NF-kB (1/1000), AKT (1/1000), p-AKT (1/1000), SAPK/JNK (1/1000), p-SAPK/JNK (1/1000), LAMC2 (1/500) were diluted in primary antibody diluent (Beyotime, Shanghai, China) according to corresponding manufacturers’ instructions. Antibodies’ information for western blotting is listed in Appendix A.

### 2.15. Cell Count Kit-8 (CCK-8) Assay

After transfection for 48 h, cells were seeded into 96-well plates at 5 × 10^3^ cells/well density and incubated in a 37 °C humidified atmosphere containing 5% CO_2_. When incubated for 24 h, 48 h, 72 h, and 96 h, the culture medium was discarded, and subsequently, 10 μL of CCK-8 Cell Counting Kit (A311-02, Vazyme, Nangjing, China) and 90 μL DMEM medium containing 10% FBS were added to each well and incubated in an incubator at 37 °C for 2–3 h before the measure at 450 nm with a SpectraMax^®^ Absorbance Reader CMax Plus (Molecular Devices, Silicon Valley, CA, USA).

### 2.16. Cell Scratch Assay

Cells were seeded in six-well plates at 8 × 10^5^ cells/well density and incubated 24 h, followed by transfection for 48 h. When cells were incubated to about 90–100% confluence, a sterile plastic micropipette tip was used to make artificial scratches, and the cells were washed 3 times with 1× PBS to remove the floating cells, followed by incubated with serum-free medium for 48 h. The images of each artificial scratch at 0 h and 48 h were acquired with microscopy at 100× magnification. The migration distance of cells was calculated by the distance of each scratch at 0 h and 48 h. The results were expressed as relative migration distances.

### 2.17. Transwell Migration Assay

After transfection for 48 h, cells were harvested and diluted in serum-free DMEM medium. Subsequently, cells (5 × 10^4^, 100 uL) were seeded into the upper chamber (8.0 μm pore size, corning Life Sciences, Corning, NY, USA), with the lower chamber filled with 600 μL DMEM medium containing 20% FBS. After incubation for 48 h, the migrated cells attached on the underside of the membrane were fixed with 4% paraformaldehyde for 20 min and stained with 0.1% crystal violet for 20 min. The cells above the membrane were wiped with a cotton ball. The migrated cells attached on the underside of the membrane were imaged with an inverted microscope (Leica, Wetzlar, Hessen, Germany) and the total cell numbers were quantified from three random fields at 100× magnification. The results were expressed as relative migration cells.

### 2.18. Cell Adhesion Assay

Matrigel (100 μL) (356234, corning Life Sciences, Corning, NY, USA) was diluted in 900 μL serum-free DMEM medium. Ninety six-well plates were pre-coated with matrigel solution (50 μL/well) and air-dried in incubator at 37 °C overnight and then washed 3 times with serum-free DMEM medium containing 0.1% bovine serum albumin (BSA) (Solarbio, Beijing, China) and blocked for 1 h with serum-free DMEM medium containing 0.5% BSA. After transfection for 72 h, cells (10^4^) were seeded with 100 uL DMEM medium containing 10% FBS into each well of the pre-coated 96-well plate and incubated for 40 min, followed by gently washing 3 times with 1× PBS. Subsequently, CCK-8 assay was performed to detect the number of adhered cells. The results were expressed as the relative number of adhesive cells.

### 2.19. Co-Immunoprecipitation (Co-IP), Liquid Chromatography Tandem Mass Spectrometry (LC-MS) and Protein–Protein Interaction (PPI) Analysis

Cells were transfected with plasmids (concentration: 2.5 ng/uL) for 72 h. Cell lysates were prepared by using IP lysis buffers (Beyotime, Shanghai, China) and further diluted in 1×PBS to a concentration of 1 μg/uL. Subsequently, 500 ug of the total protein samples were incubated with 5 μg mouse anti-FLAG monoclonal antibody and 50 μL BeyoMagTM Protein A+G Magnetic beads (P2108, Beyotime, Shanghai, China) at 4 °C for 24 h. The immunoprecipitants were washed 3 times with cold 1× PBS and dissolved with 50 μL 1× loading buffer, followed by boiling at 100 °C for 10 min. Western blotting assay was performed according to the protocol above. In addition, 30 μL of the immunoprecipitant samples were sent to Wininnovate Bio company (Shenzhen, China) for Liquid Chromatography Tandem Mass Spectrometry (LC-MS) analysis. Briefly, the immunoprecipitant samples were analyzed by western blotting. For MS analysis, the gels were firstly cut into small pieces and decolorized with certain solution, followed by dried with 100% acetonitrile. Subsequently, the gels were digested, and the peptides were obtained from them. The peptides were detected by LC-MS on a ThermoFisher Q Exactive mass spectrometer (Thermo Fisher Scientific, Waltham, MA, USA). The LC-MS data were analyzed for protein identification and quantification by using PEAKS Studio 8.5 (Bioinformatics Solutions Inc., Waterloo, ON, Canada). The local false discovery rate at PSM was 1.0% after searching against Homo sapiens database with a maximum of two missed cleavages. The following settings were selected: Oxidation (M), Acetylation (Protein N-term), Deamidation (NQ), Pyro-glu from E, Pyro-glu from Q for variable modifications as well as fixed Carbamidomethylation of cysteine. Precursor and fragment mass tolerance were set to 10 ppm and 0.05 Da, respectively. Proteins identified from LC-MS results were submitted to STRING database (STRING, https://cn.string-db.org, accessed on 11 September 2022) for PPI analysis to acquire the proteins interacting with LAMB3, followed by visualized with the Cytoscape software (version 3.9.1, https://cytoscape.org/, accessed on 2 October 2022). For endogenous protein interaction between LAMB3 and LAMC2, 500 ug of the total protein sample derived from PANC-1 or AsPC-1 cells were incubated with 1 ug rabbit anti-LAMB3 polyclonal antibody and 20 uL BeyoMagTM Protein A+G Magnetic beads at 4 °C for 24 h. Antibodies’ information for Co-IP is listed in Appendix A.

### 2.20. Cell Immunofluorescence Assay

After transfected with a plasmid containing full-length LAMB3 cDNA (pLAMB3) at 2.5 ng/uL concentration for 48 h, PANC-1 cells were seeded into dishes (801002-1, NEST, Wuxi, China) at 2 × 10^5^ cells/dish density and incubated for 48 h. After fixed with 4% paraformaldehyde for 15 min, permeabilized in immunostaining permeabilization buffers (Beyotime, Shanghai, China) for 15 min, and blocked with 5% BSA for 1 h at room temperature, the cells were added with primary antibodies against LAMC2 and Flag at 1:250 dilution in 1× PBS and incubated overnight at 4 °C. The next morning, after washed 3 times with cold 1× PBS, the cells were added with Alexa Fluor^®^488, and Alexa Fluor^TM^594 conjugated secondary antibodies at a 1:300 dilution in 1× PBS and further incubated at room temperature for 1 h in a dark condition. After washed 3 times with cold 1× PBS, the cells were incubated with DAPI at 1:1000 dilution in 1× PBS for 1 min in a dark condition, followed by washed with cold 1× PBS for 5 min and added with 100 uL fluorescence anti-fade reagent (Boster, Wuhan, China). Images were taken with a Laser confocal microscope (LSM880, Zeiss, Oberkochen, Baden-Wurtberg, Germany) at 600× magnification. Antibodies’ information for cell immunofluorescence is listed in Appendix A.

### 2.21. Mouse Pancreatic Orthotopic Xenograft Model and In Vivo Imaging Analysis

BALB/c nude mice (female, 4–6 weeks) were purchased from Guangdong Medical Laboratory Animal Center (www.gdmlac.com.cn, accessed on 7 July 2022). All animal experiments were approved by the Experimental Animal Ethics Committee of the Fifth Affiliated Hospital of Sun Yat-sen University. After anesthesia, the abdominal cavity of mice was opened, and the pancreas was exposed, then S100A10 stably knocked-down (shS100A10) and negative control (shNC) PANC-1 cells (2 × 10^6^) were injected with 10 uL 1×PBS into the tissue under the pancreatic capsule by using a micro syringe. The weight of mice was measured weekly. Two months after inoculation, in vivo imaging analysis was performed to detect the bioluminescence signals of pancreatic orthotopic tumors by using appropriate amount of VivoGlo^TM^ Luciferin (150 mg/kg) (P1042, Promega, Madison, WI, USA) with an IVIS Lumina III device (PerkinElmer, Waltham, MA, USA). Subsequently, the mice were sacrificed, and their pancreatic tumor tissues were dissected, followed by fixed with 4% paraformaldehyde. The tumor tissue sections were used for H&E staining or Immunohistochemistry. H&E staining was used for the routine histopathological examination.

### 2.22. Statistical Analysis

All analyses were conducted by using GraphPad Prism 8.0 software (GraphPad Software, San Diego, CA, USA). Mann–Whitney test was used to compare the protein expression level of interesting genes in human PDAC specimens. The correlation between target proteins in human PDAC specimens were analyzed with Spearman’s *p* statistic. Data from qRT-PCR, cell experiments and animal experiments were analyzed by Student’s *t*-test, and the results are represented as mean ± standard deviation (SD) of three independent experiments. *p* < 0.05 was considered statistically significant.

## 3. Results

### 3.1. S100A10 Is Up-Regulated in PDAC Tissues and Associated with an Unfavorable Prognosis by Integrated Bioinformatic and Human PDAC Tissue Samples Analyses

To assess the mRNA expression levels of S100s in PDAC, we examined 21 members of S100s in the Oncomine database and compared their expression levels between cancer and normal samples (Appendix A). We found that the mRNA expression levels of S100A2/4/6/10/11/13/14/16 and S100P were significantly up-regulated in PDAC tissues. Subsequently, the mRNA expression levels of the 21 genes were also analyzed in the GEPIA database. The results showed that S100A2/3/4/6/8/9/10/11/13/14/16, S100B, and S100P were over-expressed in PDAC tissues (Appendix A). After the intersection between the significantly up-regulated genes identified in these two databases, nine genes, including S100A2/4/6/10/11/13/14/16 and S100P, were noted (Figure 1A). The overall survival (OS) and recurrence-free survival (RFS) analyses for the nine genes were performed in Kaplan–Meier plotter database, and we found that only the expression levels of S100A2/10/14/16 were negatively related to the OS of PDAC patients (logrank *p* < 0.05) (Figure 1B, Appendix A), while only the expression levels of S100A2/10/11 were negatively associated with the RFS of PDAC patients (logrank *p* < 0.05) (Figure 1C and Appendix A). As the high expression levels of S100A2/10 both indicated worse RFS and worse OS in PDAC patients, the correlation analyses between the expression levels of S100A2/10 and tumor grade/stage in PDAC patients were further performed in the UALCAN database, and we found that the expression level of S100A10 was not only correlated with tumor grade (*p* < 0.05) (Figure 1D), but also tumor stage (*p* = 0.0015) (Figure 1E), while S100A2 was only associated with tumor grade (*p* < 0.05) (Appendix A). Therefore, S100A10 was taken into consideration in the following analyses. In the Oncomine database, the mRNA expression level of S100A10 was further analyzed in five datasets, and the results showed that the mRNA expression level of S100A10 was significantly higher in PDAC tissues than that in normal pancreas tissues (Appendix A). In addition, a higher S100A10 protein expression was also detected in PDAC tissues than that in pancreas non-tumor tissues in the HPA database (Appendix A) and human PDAC specimens (Figure 1F).

### 3.2. S100A10 Promotes PDAC Cells Proliferation, Migration and Adhesion

In order to elucidate the role of S100A10 in PDAC, the protein expression level of S100A10 was detected in six PDAC cancer cell lines by western blotting (Appendix A), and PANC-1 and AsPC-1 with the relatively high expression level of S100A10 were selected for further study. In vitro, the expression level of S100A10 was successfully knocked down by using two siRNAs against S100A10 and over-expressed by using a plasmid containing the full-length cDNA of S100A10 in PANC-1 and AsPC-1 cells (Figure 2A,B). Subsequently, cell migration was evaluated by using cell scratch and transwell migration assays, while cell proliferation and adhesion were evaluated by using CCK-8 assay and cell adhesion assay, respectively. The results of in vitro assays showed that knockdown of S100A10 significantly inhibited the proliferation (Figure 2C), migration (Figure 2E), and adhesion (Figure 2F) of PDAC cell lines, while over-expression of S100A10 promoted PDAC cells proliferation (Figure 2D), migration (Figure 2E) and adhesion (Figure 2F).

To further assess the tumor growth of S100A10 suppression, PANC-1 cells with stable knockdown of S100A10 (shS100A10) and corresponding negative control (shNC) were generated and used for an orthotopic xenograft model (Figure 3A,B). In this assessment, we found that the size of orthotopic pancreatic tumors in the shS100A10 group was significantly smaller than those in the shNC control group (Figure 3C–E), and the protein expression levels of S100A10 and Ki-67 were significantly inhibited in the tumor tissues of shS100A10 group (Figure 3F).

### 3.3. ITGA2 and LAMB3 Are Activated by S100A10 in PDAC Cells

As the above in vivo and vitro experiments demonstrated, S100A10 was involved in the progression of PDAC, and further analyses were performed to search for the genes activated or inhibited by S100A10 in PDAC. Firstly, 8386 and 2337 genes significantly associated with S100A10 were identified in the QCMG project (*q*-value < 0.05) and TCGA project (*q*-value < 0.01), respectively. Subsequently, 274 significant DEGs were identified in the GSE62452 dataset (*p*adj-value < 0.05, |log2-fold change| > 1), and 2926 significant DEGs were identified in PDAC by integrated bioinformatic analysis on TCGA combined with GTEx (*p*adj-value < 0.05, |log2-fold change| > 2). After the intersection of these genes obtained from the above four analyses, 55 genes were identified to be not only significantly associated with the expression of S100A10 but also significantly differentially expressed in PDAC tissues (Figure 4A). Survival analyses for these 55 genes were further performed in the GEPIA database, and we found that only 38 of these genes were identified to be significantly associated with the OS of PDAC patients (logrank *p*-value < 0.05) (Appendix A). The results of GO and KEGG pathway enrichment analyses indicated that these 38 genes were mainly enriched in PI3K/AKT, extracellular matrix, and cell-adhesion-related pathways (Figure 4B,C). In addition, RNAseq was also performed to help identify the genes activated or inhibited by S100A10 in PDAC, and 134 genes were identified and further used for KEGG pathway enrichment analysis in the DAVID database. The results showed that these 134 genes were mainly enriched in the pathways related to focal adhesion (*p*-value = 0.0022), human papillomavirus infection (*p*-value = 0.0063), pathways in cancer (*p*-value = 0.0082), viral carcinogenesis (*p*-value = 0.0115), Cushing syndrome (*p*-value = 0.0199), Small cell lung cancer (*p*-value = 0.0236), axon regeneration (*p*-value = 0.0308) and NF-kappa B signaling pathway (*p*-value = 0.0324) (Figure 4D). As the *p*-value of the focal adhesion pathway is the smallest, the genes related to the focal adhesion pathway were noted (Figure 4E) and further intersected with the 38 genes identified in the analyses of the public database (Figure 4F). Finally, ITGA2 and LAMB3 were regarded as the potential genes activated by S100A10 in PDAC.

In the public online databases analyses, the expression levels of ITGA2 and LAMB3 were not only significantly up-regulated in PDAC tissues (Appendix A–C), but also negatively related to the OS (logrank *p* < 0.05) (Appendix A) and RFS (logrank *p* < 0.05) (Appendix A) of PDAC patients as well as correlated with the expression level of S100A10 in PDAC tissues (ITGA2 vs. S100A10: Person R = 0.42, *p*-value = 3.3 × 10^9^; LAMB3 vs. S100A10: Person R = 0.68, *p*-value = 0) (Appendix A). The results of western blotting showed that the protein expression levels of ITGA2 and LAMB3 were also inhibited or activated when the expression level of S100A10 was knocked down or over-expressed in PDAC cell lines (Figure 5A). In addition, the results of IHC staining on mice pancreatic tumor tissues from the orthotopic xenograft model also indicated that ITGA2 and LAMB3 were significantly inhibited in the shS100A10 group, compared with that in the shNC control group (Figure 5B). In the human PDAC specimen analyses, the IHC staining showed that both the protein expression levels of ITGA2 and LAMB3 were not only significantly up-regulated in PDAC tissues (Figure 5C–E) but also positively correlated with the protein expression level of S100A10 in PDAC tissues (ITGA2 vs. S100A10: Spearman R = 0.4409, *p*-value = 0.0056, *n* = 38; LAMB3 vs. S100A10: Spearman R = 0.3444, *p*-value = 0.0342, *n* = 38) (Figure 5F,G).

### 3.4. LAMB3 Promotes PDAC Cells Proliferation, Migration and Adhesion

As LAMB3 has also been identified as a prognostic biomarker for predicting the progression of PDAC patients by integrated bioinformatic analysis in the latest study [15], and its molecular mechanism in PDAC is still unknown, LAMB3 was selected for further study. The expression level of LAMB3 was successfully knocked down with a siRNA against LAMB3 and over-expressed with a plasmid containing the full-length cDNA of LAMB3 in PANC-1 and AsPC-1 cells (Figure 6A,B), followed by used for in vitro function experiments. The results of CCK-8, cell scratch, cell transwell migration, and cell adhesion assays indicated that knockdown of LAMB3 inhibited the proliferation (Figure 6C), migration (Figure 6E), and adhesion (Figure 6F) of PDAC cell lines, while over-expression of LAMB3 exerted an opposite effect on cell proliferation (Figure 6D), migration (Figure 6E) and adhesion (Figure 6F). In addition, in vitro rescue experiments were also performed to further validate the role of LAMB3 in the progression of PDAC. After the stable knockdown of S100A10 (shS100A10 group) in PANC-1 and AsPC-1 cells, the protein expression level of LAMB3 was inhibited, and subsequently, LAMB3 was re-expressed in the shS100A10-pLAMB3 group by using a plasmid containing full-length cDNA of LAMB3 (Figure 7A). The results of rescue assays displayed that cell proliferation (Figure 7B), migration (Figure 7C–E), and adhesion (Figure 7F) were significantly inhibited in PDAC cell lines in the shS100A10 group, while the inhibitory effect was reversed with the re-expression of LAMB3 in the shS100A10-pLAMB3 group.

### 3.5. S100A10 Activates LAMB3 through JNK Pathway

LAMB3 has been proven to be activated by S100A10. However, the molecular mechanism of S100A10 activating LAMB3 is not elucidated. First of all, we performed a Co-IP assay, and the results showed that S100A10 does not directly interact with LAMB3 (Appendix A–C). Subsequently, the protein expression levels of classical signaling pathways, including NF-kB, AKT, and JNK, were detected in the control (shNC) and S100A10 stably knocked-down (shS100A10) PANC-1 and AsPC-1 cells, and we found that the protein expression of phosphorylated JNK (p-JNK) was significantly inhibited in the shS100A10 group in both PDAC cell lines (Figure 8A). A similar change was observed in the expression of p-JNK in both PDAC cell lines transfected with siRNAs against S100A10, while the expression of p-JNK was activated in both PDAC cell lines transfected with a plasmid to over-express S100A10 (Figure 8B). To further investigate whether S100A10 activated LAMB3 by the JNK pathway, a JNK inhibitor (SP600125) and a siRNA against JNK1 (siJNK1) were used. As shown in Figure 8C,D, LAMB3 was activated after the over-expression of S100A10 in the pS100A10 group, and its activation was reversed by inhibiting the expression of JNK with SP600125 or siJNK1, indicating that S100A10 activates LAMB3 by JNK pathway in PDAC cells.

### 3.6. LAMB3 Interacts with LAMC2 in PDAC Cells

In order to identify the proteins that interact with LAMB3 in PDAC, Co-IP was performed after the over-expression of LAMB3 (pLAMB3) in PANC-1 cells, and the protein samples were further used for LC-MS analysis. The results of LC-MS analysis indicated that 394 proteins were identified in the control (pNC) group, while 404 proteins were identified in the pLAMB3 group. After the intersection of these proteins identified in these two groups, 107 proteins were identified as the potential target proteins interacting with LAMB3 (Figure 8E). Subsequently, the 107 proteins and LAMB3 were imported to the STRING database for PPI analysis, and the results showed that LAMC2 and KRT17 were predicted to interact with LAMB3 directly (Figure 8F), while the PPI network constructed by LAMB3 in STRING database indicated the interaction of LAMB3 with several proteins including LAMC2 (Figure 8G). The results of bioinformatics analyses in the GEPIA and HPA databases showed that the expression of LAMC2 was not only significantly up-regulated in PDAC tissues (Appendix A) but also showed a strong correlation with the expression of LAMB3 (Pearson R = 0.74, *p*-value = 0) in PDAC (Appendix A). In addition, we also verified the direct interaction between LAMB3 and LAMC2 (Figure 8H), and these two proteins co-localized in the cytoplasm of PDAC cells (Figure 8I).

## 4. Discussion

S100A10 has been demonstrated to have a certain value in the diagnosis, treatment, and prognosis assessment of a variety of cancers since being first identified in 1985 [16]. In lung squamous cell carcinoma, the expression of S100A10 in tumor tissue was positively correlated with tumor size, TMN stage, and lymphatic metastasis [8], while elevated S100A10 in lung adenocarcinoma indicated poorer tumor differentiation, higher TMN stage, more frequent intratumoral vascular invasion and a poorer prognosis [17]. What is more, the serum S100A10 levels in lung cancer were not only significantly higher than that in patients with benign lung nodules and healthy cancer-free controls but also positively associated with TNM stage and lymphatic metastasis [18], indicating that S100A10 is not only a prognostic marker but also a potential diagnostic maker for lung cancer. Similarly, Yi Tan et al. reported that S100A10 might serve as a potential biomarker for early detection and a potential therapeutic target for gallbladder cancer [9]. In addition, higher S100A10 expression is also reported to be correlated with shorter survival rates in gastric cancer [19] and breast cancer [7]. In the present study, S100A10 was demonstrated to be significantly up-regulated in PDAC tissues by integrated bioinformatic and human PDAC sample analyses, and high S100A10 expression is associated with advanced clinical stage, poorer differentiation, and shorter survival rate. It is implied that S1000A10 has the potential to be a prognostic biomarker for PDAC patients.

Sustained proliferative signaling is one of the hallmarks of cancer. It is reported that decreased S100A10 expression not only inhibits ovarian cancer cell proliferation and colony formation in vitro but also remarkably suppresses ovary tumor growth in vivo [20]. Likewise, the knockdown of S100A10 significantly reduced the proliferation and metastasis capacity of colorectal cancer cells [10]. In hepatocellular carcinoma, the over-expression of S100A10 accelerates cell proliferation in Hep3B and Huh-7 cells, while decreased S100A10 expression shows the opposite effect on the proliferation capacity of SK-Hep-1 and HepG2 cells [11]. Additionally, it is demonstrated that S100A10 promotes the malignant growth of cancer cells by activating the AKT/mTOR signaling pathway in osteosarcoma [21] and the Src/ANXA2/AKT/mTOR signaling pathway in gastric cancer [19]. Collectively, S100A10 serves as a positive regulator in cell proliferation in multiple cancers. In this study, we revealed that S100A10 not only promotes PDAC cell proliferation in vitro but also accelerates pancreatic tumor growth in vivo, which is consistent with the in vivo results in a previous study [14]. However, Moamen Bydoun et al. reported that similar proliferation rates were observed between S100A10 stably depleted PANC-1 cells and scramble control cells in vitro [14], which is inconsistent with our in vitro results. Further studies are needed to elucidate the effect of S100A10 on cell proliferation in vitro.

Activation of invasion and metastasis is another hallmark of cancer. S100A10 has been reported to promote tumor progression and metastasis by regulating multiple signaling pathways related to cell migration, invasion, metastasis, and angiogenesis [22]. S100A10 is known for its role in interacting with ANXA2 to form the ANXA2/S100A10 heterotetrameric complex (AIIt), which is identified as one of the oncogenic plasminogen receptors [23]. The activation of plasminogen by AIIt not only activates pro-matrix metalloproteinases (pro-MMPs) but also induces the angiogenesis of cancer stroma via ECM-associated proangiogenic growth factors, which both play an important role in the invasion and metastasis of cancer [7]. Choi KS et al. [24] reported that the depletion of S100A10 significantly reduced the lung metastases of fibrosarcoma cells in vivo, while Zhang et al. [25] revealed that stable suppression of S100A10 in colorectal adenocarcinoma CCL-222 cells without ANXA2 expression resulted in a complete loss of plasminogen-dependent invasiveness. It is indicated that S100A10 may promote the invasiveness and metastasis of these cancer cells through an ANXA2-independent pathway. In addition, S100A10 also plays an important role in cell migration. Kyle D. Phipps et al. [26] reported that S100A10 is crucial for tumor-promoting macrophage migration to tumor sites. As a regulator of the filamentous actin network, the depletion of S100A10 impairs cell motility and further inhibits cell migration in squamous carcinoma A431 cells [27]. Our results reveal that S100A10 enhances the migration capacity of PDAC cell lines in vitro. In addition, the genes significantly associated with S100A10 and significantly differentially expressed in PDAC are enriched in the cellular components, including cortical actin cytoskeleton and cortical cytoskeleton. It is reported that S100A10 participates in the organization of actin stress fibers and the formation of focal adhesions by activating Rac1, promoting cell migration and spreading [28]. Furthermore, SUMO1-modified S100A10 can translocate into nuclei to up-regulate the expression of genes associated with actin dynamics and cytoskeleton remodeling, followed by the enhanced migration capacity of polyploid tumor giant cells and their daughter cells [29]. Based on these results, we speculated that S100A10 might participate in the cytoskeleton remodeling of PDAC cells and thus enhance cell motility and migration ability.

Focal adhesion is an integrin-based structure that mediates the adhesion of cells to ECM and involves in the signaling transmission between cytoplasm and ECM, playing an important role in cell proliferation, adhesion, migration, and invasion [30]. We also found that the genes (including LAMB3, LAMC2, ITGA2, FLNA, EGFR, TLN2, and VCL) activated by S100A10 in PDAC were mainly enriched in the focal adhesion pathway by RNAseq and KEGG pathway enrichment analyses. Subsequently, ITGA2 and LAMB3 were further proved to be activated by S100A10 in PDAC, which both validated to be not only up-expressed in PDAC tissues but also associated with the prognosis of PDAC patients [31]. In addition, ITGA2 has also been demonstrated to promote PDAC progression via the focal adhesion pathway [32], and LAMB3 is also identified as an oncogene related to the focal adhesion pathway in cervical squamous cell carcinoma [33]. In vitro, S100A10 was further demonstrated to enhance the adhesion capacity of PDAC cells to the extracellular matrix in our study. In a previous study, the activation of focal adhesion signaling pathways has been reported to promote cell adhesion, migration, and invasion in breast cancer [34]. As a signal transducer between ECM and cytoskeleton, focal adhesion also influences cytoskeleton remodeling and cell motility in cancer [35,36]. Therefore, It is indicated that the enhanced cell adhesion may be one of the reasons why S100A10 promotes PDAC cell migration in vitro.

LAMB3 is emerging as a potential therapeutic target for cancer due to its oncogenic role. LAMB3 was reported to be up-regulated in thyroid cancer and promote papillary thyroid cancer cell migration and invasion by activating the c-MET/AKT signaling pathway [37]. In HNSCC, high LAMB3 expression is not only correlated with unfavorable prognosis in HNSCC patients but also associated the cisplatin cytotoxicity to HNSCC cells [38]. In gastric cancer, the up-regulation of LAMB3 plays an important role in cancer progression [39]. Similarly, our results indicated that higher LAMB3 expression was detected in PDAC tissues than in matched pancreas normal tissues, whose expression was inversely associated with the survival rates of PDAC patients. Furthermore, LAMB3 was identified as an oncogene to promote PDAC cell proliferation, migration, and adhesion in vitro. In line with our results, LAMB3 has been proven to be up-regulated in PDAC [15,31] and enhance PDAC malignant phenotypes through PI3K/AKT axis [40]. In previous studies, LAMB3 was also reported to promote tumor progression and chemoresistance through the AKT-FOXO3/4 axis and be transcriptionally regulated by the BRD2/acetylated ELK4 complex [41] and polymeric immunoglobulin receptor [42] in colorectal cancer. In the present study, we also found that LAMB3 was activated by S100A10 in PDAC cells, and its expression was positively correlated with S100A10 expression in human PDAC tissues. Additionally, S100A10-mediated expression of LAMB3 was dependent on the JNK pathway, while LAMB3 has been reported to be modulated by miR-24-3p to influence the progression of PDAC [43]. Collectively, S100A10 promoted the progression of PDAC via JNK/LAMB3 axis.

Laminin-332 (LM-332) is an important adhesive molecule of basement membranes and is involved in regulating multiple biological functions related to cell adhesion, migration, and invasion in normal tissues [44]. As a unique subunit of LM-332, LAMC2 has been reported to be up-regulated in esophageal squamous cell carcinoma [45] and ovarian carcinoma [46]. In PDAC, the elevated expression of LAMC2 has been detected not only in tumor tissues [47] but also in serum [48]. In addition, LAMC2 has also been identified as an oncogene to promote PDAC progression by modulating EMT- and ATP-binding assiette transporters [49] or EGFR/ERK_1/2_/AKT/mTOR axis [50]. It is indicated that LAMC2 may be served as a candidate biomarker for PDAC diagnosis and treatment. In our study, the expression of LAMC2 was significantly up-regulated in PDAC tissues and showed a strong correlation with the expression of LAMB3, indicating a co-expression pattern of these two genes in PDAC. Consistent with our results, LAMC2 has been reported to be frequently co-expressed with LAMB3 in colorectal cancer, squamous cell carcinoma of the tongue [51], gastric cancer [39], prostate cancer [52] and PDAC [53], whose intracellular retention at the invasive front of cancer cells has been considered as a more invasive phenotype. We also revealed an interaction between LAMB3 and LAMC2 in PDAC cells, and these two proteins co-localized in the cytoplasm of PDAC cells. In previous studies, LAMB3 has been reported to interact with LAMC2 to form the β3γ2 heterodimer in the cytoplasm of colorectal cancer cells and thus contribute to cancer cell budding [54,55]. Based on these results, we postulated that the interaction between LAMB3 and LAMC2 may be a significant step in the progression of PDAC. Further studies are needed to confirm this hypothesis in the future.

## 5. Conclusions

In this study, we found that S100A10 is significantly up-regulated in PDAC tissues and identified as a poor prognostic factor for PDAC patients. Subsequently, S100A10 was demonstrated to promote PDAC cell proliferation, migration, and adhesion in vitro and accelerate pancreatic tumor growth in vivo. Mechanically, S100A10 activates LAMB3 through the JNK pathway, and LAMB3 further interacts with LAMC2 in PDAC cells, contributing to the progression of PDAC. All in all, S100A10 promotes PDAC progression through JNK/LAMB3-LAMC2 axis. Apart from these findings, there are several limitations to this study. Firstly, the role of S100A10 in PDAC metastasis needs to be further verified in vivo. In addition, LAMB3 has been confirmed to promote PDAC cell proliferation, migration, and adhesion in vitro, but whether it plays the same oncogenic role in vivo still needs further study. Finally, the interaction between LAMB3 and LAMC2 has been demonstrated in this study, but how this interaction promotes the progression of PDAC remains to be clarified in the future. Taken together, it is indicated that S100A10 has the potential to be served as a candidate biomarker for PDAC prognostic evaluation and a therapeutic target for PDAC patients.

## Figures and Tables

**Figure 1 cancers-15-00202-f001:**
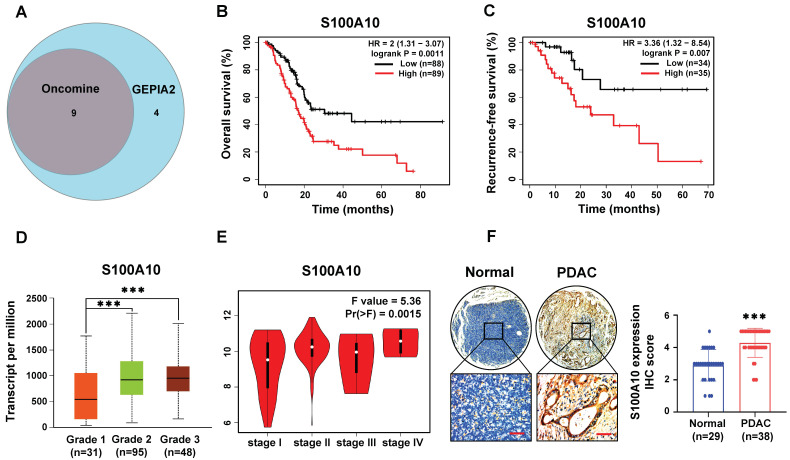
S100A10 is up-regulated in PDAC tissues and associated with an unfavorable prognosis by integrated bioinformatics and human PDAC samples analyses. (**A**) The intersection of the significantly up-regulated S100s members analyzed in Oncomine database and GEPIA database. (**B**,**C**) The overall survival and recurrence-free survival analyses for S100A10 in PDAC patients in Kaplan–Meier Plotter database (cutoff of logrank *p*-value: 0.05). (**D**,**E**) The correlation analyses between S100A10 and tumor grade/stage in PDAC patients in UALCAN database (grade: *** *p* < 0.001 by Student’s *t*-test; stage: *p* < 0.05 by F-test). (**F**) IHC staining of S100A10 in PDAC tissues and pancreatic non-tumor tissues (*** *p* < 0.001 by Mann–Whitney test). Scale bar: 50 um.

**Figure 2 cancers-15-00202-f002:**
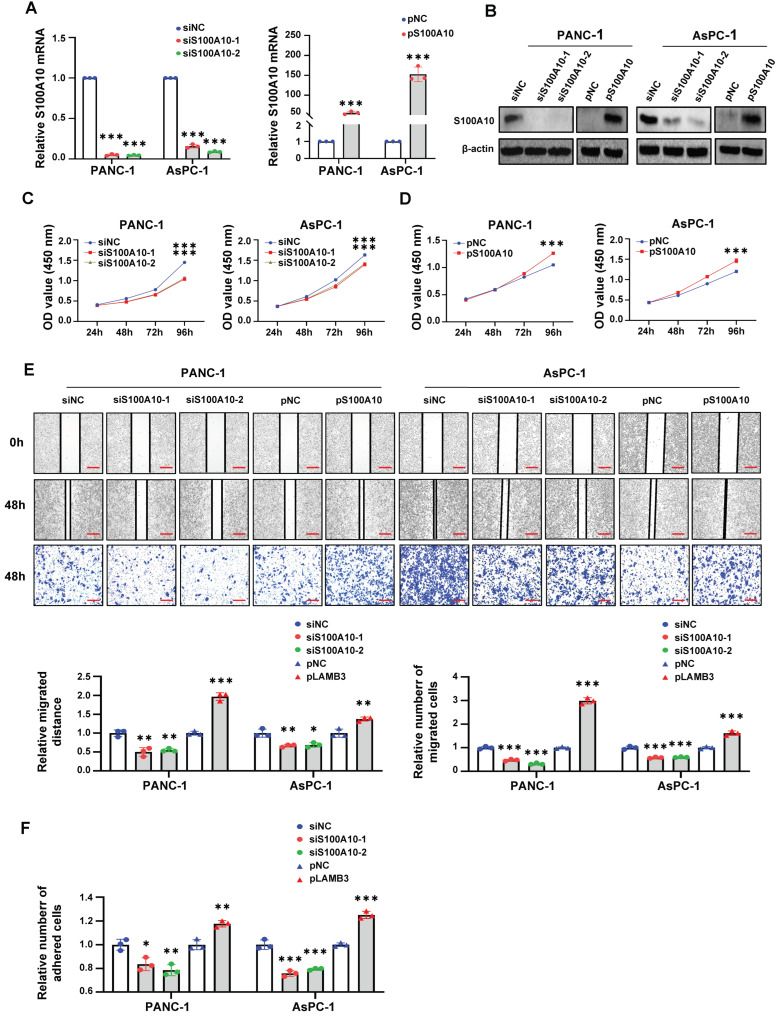
S100A10 promotes PDAC cells proliferation, migration and adhesion in vitro. (**A**,**B**) RT-qPCR and Western blotting were used to measure the expression of S100A10 in PANC-1 and AsPC-1 cells after transfection with siRNAs against S100A10 (siS100A10-1, siS100A10-2) or control siRNA (siNC), a plasmid to over-express S100A10 (pS100A10) or control plasmid (pNC) for 48 h. (**C**,**D**) After transfection with siNC, siS100A10-1, siS100A10-2, pNC or pS100A10 for 48 h, cell proliferation of PANC-1 and AsPC-1 cells were determined by CCK-8 assay at indicated time points (*n* = 3). (**E**) After transfection with siNC, siS100A10-1, siS100A10-2, pNC or pS100A10 for 48 h, cell migration of PANC-1 and AsPC-1 cells were determined by cell scratch assay and transwell migration assay at indicated time points (*n* = 3). Scale bar: 200 um. (**F**) After transfection with siNC, siS100A10-1, siS100A10-2, pNC or pS100A10 for 72 h, cell adhesion of PANC-1 and AsPC-1 cells were determined by cell adhesion assay (*n* = 3). The transfection concentrations of siRNAs and plasmids were 100 nM and 2.5 ng/uL, respectively. Values represent mean ± SD (*n* = 3). * *p* < 0.05, ** *p* < 0.01 and *** *p* < 0.001 by Student’s *t*-test. Whole western blot images are showed in Appendix A.

**Figure 3 cancers-15-00202-f003:**
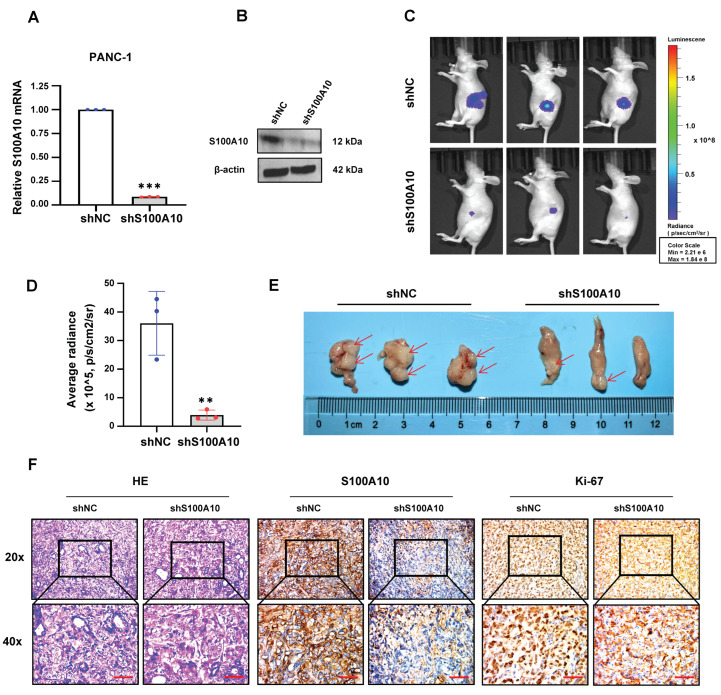
S100A10 knockdown inhibits pancreatic tumor growth in vivo. (**A**,**B**) RT-qPCR and Western blotting were used to measure the expression of S100A10 in the control (shNC) and S100A10 stably knocked-down (shS100A10) PANC-1 cells. (**C**,**D**) Luciferase images from shNC and shS100A10 PANC-1 cells orthotopic xenograft mice (*n* = 3). Luciferase images represent one picture captured and the values of average radiance are represented as mean ± SEM of the indicated number of mice. (**E**) Tumor size in shNC and shS100A10 PANC-1 cells orthotopic xenograft mice (*n* = 3). (**F**) HE stain and IHC stain of S100A10/Ki-67 in shNC and shS100A10 PANC-1 cells pancreatic tumor tissues. Scale bar: 50 um. Values represent mean ± SD (*n* = 3). ** *p* < 0.01 and *** *p* < 0.001 by Student’s *t*-test. Whole western blot images are showed in Appendix A.

**Figure 4 cancers-15-00202-f004:**
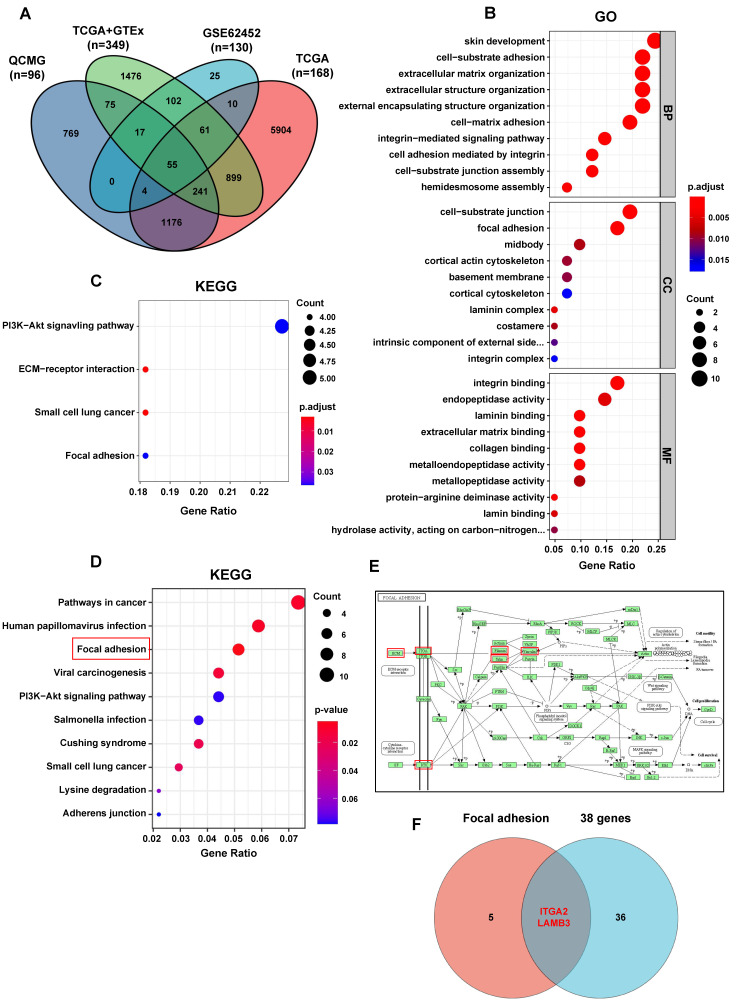
ITGA2 and LAMB3 are predicted to be activated by S100A10 in PDAC by integrated bioinformatic analyses. (**A**)The intersection between the genes significantly associated with S100A10 in QCMG/TCGA projects (QCMG project: *q*-value < 0.05, TCGA project: *q*-value < 0.01) and the significant DEGs identified in PDAC from GSE62452 dataset (*p*adj-value < 0.05, fold change > 2 or <0.5) and TCGA + GTEx database (*p*adj-value < 0.05, fold change > 4 or <0.25). (**B**,**C**) GO and KEGG pathway enrichment analyses of the 38 genes identified to be not only significantly associated with S100A10 and significantly differentially expressed in PDAC, but also significantly associated with the OS of PDAC patients in GEPIA database (logrank *p*-value < 0.05). (**D**,**E**) The KEGG pathway enrichment analysis of the 134 genes identified by RNAseq was performed in DAVID database and the focal adhesion pathway was the most significant pathway (*p*-value = 0.0022). (**F**) The intersection between the 38 genes identified above by pubic databases analyses and the seven genes enriched in the focal adhesion pathway by RNAseq analysis.

**Figure 5 cancers-15-00202-f005:**
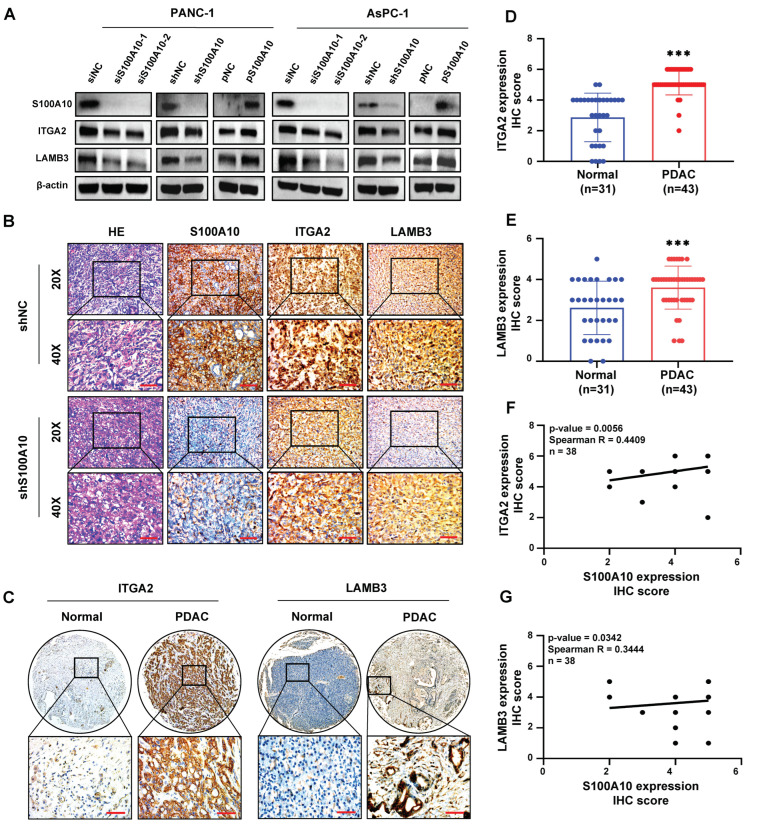
ITGA2 and LAMB3 are proved to be activated by S100A10 in PDAC. (**A**) After transfection with siRNAs against S100A10 (siS100A10-1, siS100A10-2) or control siRNA (siNC), a plasmid to over-express S100A10 (pS100A10) or control plasmid (pNC) for 72 h, western blotting was used to measure the expression of indicated proteins in the siNC, siS100A10-1, siS100A10-2, pNC, pS100A10, S100A10 stably knocked-down (shS100A10) or control (shNC) PANC-1 and AsPC-1 cells. The transfection concentrations of siRNAs and plasmids were 100 nM and 2.5 ng/uL, respectively. (**B**) HE stain and IHC stain of S100A10/ITGA2/LAMB3 in the control (shNC) and S100A10 stably knocked-down (shS100A10) PANC-1 cells pancreatic tumor tissues from orthotopic xenograft mice. Scale bar: 50 um. (**C**–**E**) IHC stain of ITGA2/LAMB3 in human PDAC tissues and pancreatic non-tumor tissues (*** *p* < 0.001 by Mann–Whitney test). Scale bar: 50 um. (**F**,**G**) The spearman correlation analyses between S100A10 and LAMB3/ITGA2 in human PDAC tissues (cutoff of *p*-value: 0.05). Whole western blot images are showed in Appendix A.

**Figure 6 cancers-15-00202-f006:**
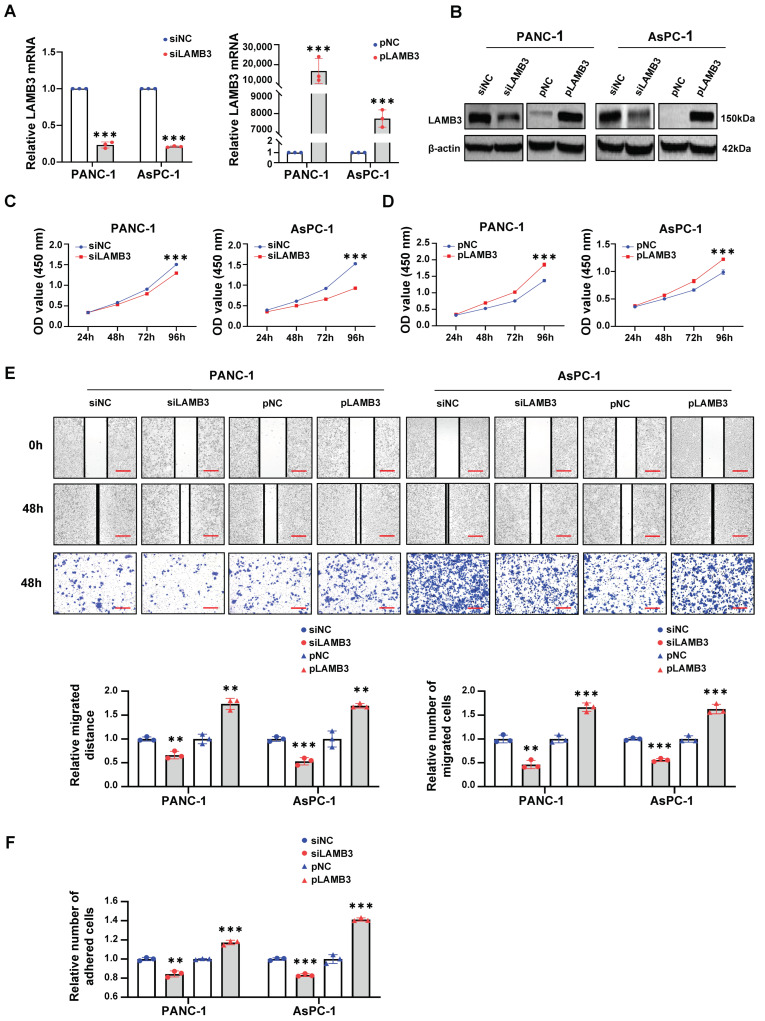
LAMB3 promotes PDAC cells proliferation, migration and adhesion in vitro. (**A**,**B**) RT-qPCR and Western blotting were used to measure the expression of LAMB3 in PANC-1 and AsPC-1 cells after transfection with a siRNA against LAMB3 (siLAMB3) or control siRNA (siNC), a plasmid to over-express LAMB3 (pLAMB3) or control plasmid (pNC) for 48 h. (**C**,**D**) After transfection with siNC, siLAMB3, pNC or pLAMB3 for 48 h, cell proliferation of PANC-1 and AsPC-1 cells were determined by CCK-8 assay at indicated time points (*n* = 3). (**E**) After transfection with siNC, siLAMB3, pNC or pLAMB3 for 48 h, cell migration of PANC-1 and AsPC-1 cells were determined by cell scratch assay and transwell migration assay at indicated time points (*n* = 3). Scale bar: 200 um. (**F**) After transfection with siNC, siLAMB3, pNC or pLAMB3 for 72 h, cell adhesion of PANC-1 and AsPC-1 cells were determined by cell adhesion assay (*n* = 3). The transfection concentrations of siRNAs and plasmids were 100 nM and 2.5 ng/uL, respectively. ** *p* < 0.01 and *** *p* < 0.001 by Student’s *t*-test. Whole western blot images are showed in Appendix A.

**Figure 7 cancers-15-00202-f007:**
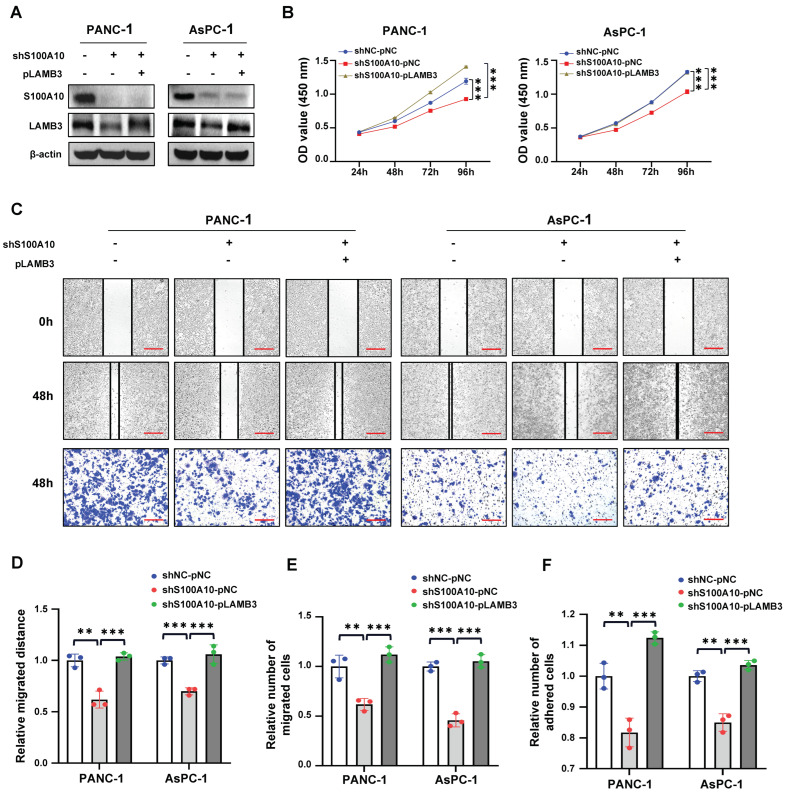
The re-expression of LAMB3 reverses the inhibitory effect of S100A10 knockdown on PDAC cells proliferation, migration and adhesion in vitro. (**A**) The S100A10 stably knocked-down (shS100A10) and control (shNC) PANC-1 and AsPC-1 cells were transfected with a plasmid to over-express LAMB3 (pLAMB3) or control plasmid (pNC) for 48 h. Western blotting was used to measure the expression of indicated proteins in the control (shNC-pNC), S100A10 stably knocked-down (shS100A10-pNC) and S100A10 stably knocked-down plus LAMB3-reexpressed (shS100A10-pLAMB3) PANC-1 and AsPC-1 cells. (**B**) After transfection with pNC or pLAMB3 for 48 h, cell proliferation was determined by CCK-8 assay in shNC-pNC, shS100A10-pNC and shS100A10-pLAMB3 PANC-1 and AsPC-1 cells at indicated time points (*n* = 3). (**C**–**E**) After transfection with pNC or pLAMB3 for 48 h, cell migration was determined by cell scratch assay and transwell migration assay in shNC-pNC, shS100A10-pNC and shS100A10-pLAMB3 PANC-1 and AsPC-1 cells at indicated time points (*n* = 3). Scale bar: 200 um. (**F**) After transfection with pNC or pLAMB3 for 72 h, cell adhesion was determined by cell adhesion assay in shNC-pNC, shS100A10-pNC and shS100A10-pLAMB3 PANC-1 and AsPC-1 cells (*n* = 3). The transfection concentration of plasmids was 1 ng/uL. Values represent mean ± SD (*n* = 3). ** *p* < 0.01 and *** *p* < 0.001 by Student’s *t*-test. Whole western blot images are showed in Appendix A.

**Figure 8 cancers-15-00202-f008:**
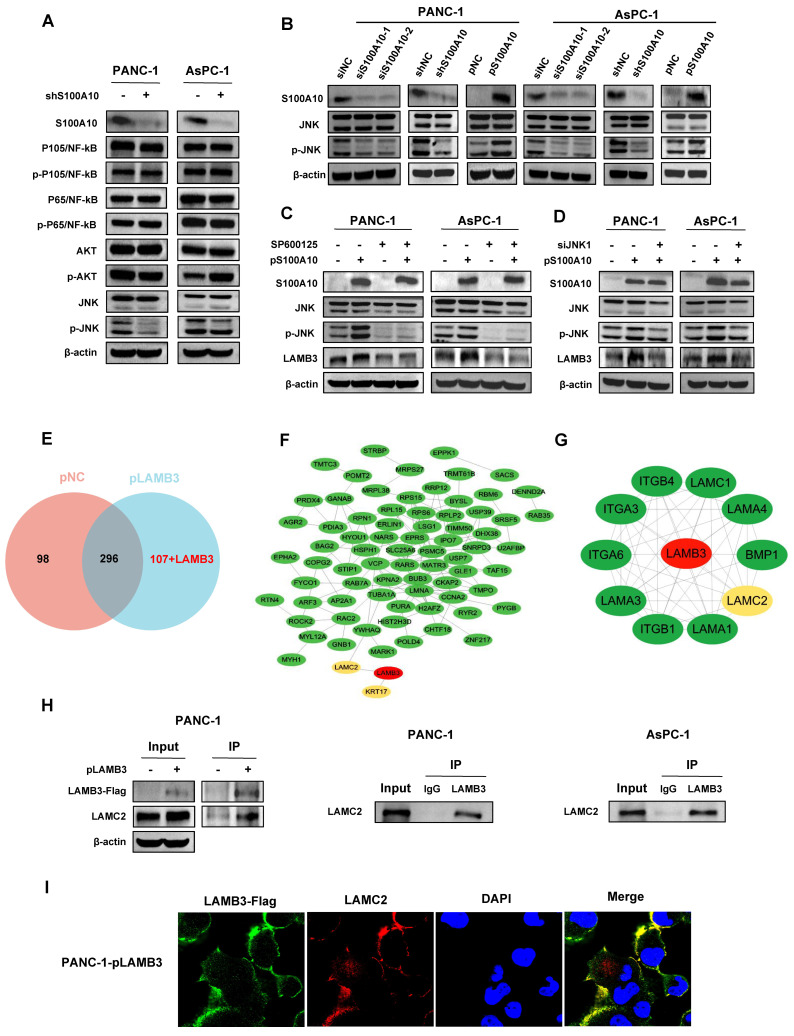
S100A10 activates LAMB3 through JNK pathway and LAMB3 further interacts with LAMC2 in PDAC cell lines. (**A**) Western blotting was used to measure the expression of indicated proteins in the control (shNC) and S100A10 stably knocked-down (shS100A10) PANC-1 and AsPC-1 cells. (**B**) After transfection with siRNAs against S100A10 (siS100A10-1, siS100A10-2) or control siRNA (siNC), a plasmid to over-express S100A10 (pS100A10) or control plasmid (pNC) for 72 h, western blotting was used to measure the expression of indicated proteins in siNC, siS100A10-1, siS100A10-2, pNC, pS100A10, shNC or shS100A10 PANC-1 and AsPC-1 cells. (**C**) After transfection with pNC or pS100A10 for 24 h, PANC-1 and AsPC-1 cells were treated with a JNK inhibitor (SP600125) at 60 umol concentration for 48 h. Western blotting was used to measure the expression of indicated proteins in pNC, pS100A10, SP600125-treated or pS100A10 plus SP600125-treated PANC-1 and AsPC-1 cells. (**D**) After transfection with pNC or pS100A10 for 24 h, PANC-1 and AsPC-1 cells were transfected with a siRNA against JNK1 (siJNK1) or control siRNA (siNC) for 48 h. Western blotting was used to measure the expression of indicated proteins in the control, pS100A10 or pS100A10 plus siJNK1 PANC-1 and AsPC-1 cells. (**E**) The intersection of the proteins pulled down in the control (pNC) and LAMB3-overexpressed (pLAMB3) PANC-1 cells by Co-IP and LC-MS analyses. (**F**) The PPI analysis of the 108 proteins identified to be only pulled down in the pLAMB3 PANC-1 cells in STRING database. (**G**) The PPI network constructed by LAMB3 in STRING database. (**H**) Co-IP was used to determine the interaction between LAMB3 and LAMC2 in PANC-1 and AsPC-1 cells. (**I**) Cell immunofluorescence assay was used to determine the location of LAMB3 and LAMC2 in PANC-1 cells transfected with pLAMB3 for 48 h. The transfection concentrations of siRNAs and plasmids were 100 nM and 2.5 ng/uL, respectively. Whole western blot images are showed in Appendix A.

## Data Availability

For all data requests, please contact the corresponding authors.

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
