# Peer review of "S100A10 Promotes Pancreatic Ductal Adenocarcinoma Cells Proliferation, Migration and Adhesion through JNK/LAMB3-LAMC2 Axis"

_cancers, 2022, doi:10.3390/cancers15010202_

Round 1

Reviewer 1 Report

The manuscript "S100A10 Promotes Pancreatic Cancer Cells Proliferation, Migra- 2 tion and Adhesion through JNK/LAMB3-LAMC2 Axis" adds knowledge to the field and presents potentially interesting findings. Nevertheless, some questions should be addressed in order to improve its scientific quality:

- After the proposal of S100A10 as a novel biomarker in pancreatic ductal adenocarcinoma (https://www.ncbi.nlm.nih.gov/pmc/articles/PMC6210040/), the authors have made an effort to build knowlegde on that and perfom mechanistical assays. Nevertheless, Methods are poorly described and should be more extended i.e. 5. Human Protein Atlas database (HPA) Analysis does not even describe immunohistochemistry.

- Regarding immunohistochemistry, the resolution of Figure 1F and 3F and all immunohistochemistry figures should be improved

- Results should be more clearly stated, without any misleading statements (i.e. avoid the use of "regulation" or "downstream" and instead use "activate" or "inhibit" to give specific information. for example 3.3. ITGA2 and LAMB3 Are the Downstream Genes of S100A10 in Pancreatic Cancer Cells)

Reviewer 2 Report

Here, Lin and Zeng describe that the S100A10 protein promotes the proliferation of pancreatic ductal adenocarcinoma cells and adhesion through JNK/LAMB3-LAMC2. They examined the hypothesis bioinformatically by the use of public databases and by in vitro and in vivo xenograft studies. Although the manuscript and figures are well prepared, I have some concerns:

Major concerns:

1. Fig. 2E, 6E and 7E: grey scale images should be improved because a difference between the images is hard to detect. Brightness and contrast should be increased with the same parameters for all images.

2. The authors examined pancreatic ductal adenocarcinoma and not pancreatic cancer in general. Therefore, the text should be revised and the abbreviation PDAC should be used — PC is the abbreviation for prostate cancer.

3. Materials, section 2.20 please revise typing error in title.

4. Figure Legends: should be improved, e.g. provide information about the abbreviations used in the figures, e.g. siNC: XY, si100A10-1: XY, provide concentrations of siRNAs and timepoints of incubation.

5. The new finding that S100A10 promotes progression of PDAC should be included into the standard nomogram for PDAC, that is based on clinicopathological parameters. The authors should examine bioinformatically that insertion of S100A10 improves the significance of prediction of overall survival. Finally, the authors should make this new improved nomogram online available for use by clinicians and patients to calculate prognosis.

Minor concern:

Language revision by a native speaker is required.

Round 2

Reviewer 2 Report

All my critic points have been addressed